# Significance of two transmembrane ion gradients for human erythrocyte volume stabilization

**F. I. Ataullakhanov[1,2,3,4,5], M. V. Martinov[1], Qiang Shi[6,7], V. M. Vitvitsky[1]** *

**1** Center for Theoretical Problems of Physico-Chemical Pharmacology, RAS, Moscow, Russia, **2** M.V. Lomonosov Moscow State University, Moscow, Russia, **3** Moscow Institute of Physics and Technology, Moscow Region, Dolgoprudny, Russia, **4** Dmitry Rogachev National Medical Research Center for Pediatric Hematology, Oncology and Immunology, Moscow, Russia, **5** University of Pennsylvania (UPENN), Philadelphia, PA, United States of America, **6** State Key Laboratory of Polymer Physics and Chemistry, Changchun Institute of Applied Chemistry, Chinese Academy of Sciences, Changchun, Jilin, China, **7** University of Science and Technology of China, Hefei, Anhui, China

* victor_vitvitsky@yahoo.com

**Data Availability Statement:** All relevant data are within the paper.

**Funding:** This work was supported in part by grant № 21-45-00012 from the Russian Science

## Abstract

Functional effectiveness of erythrocytes depends on their high deformability that allows them to pass through narrow tissue capillaries. The erythrocytes can deform easily due to discoid shape provided by the stabilization of an optimal cell volume at a given cell surface area. We used mathematical simulation to study the role of transport Na/K-ATPase and transmembrane $Na^+$ and $K^+$ gradients in human erythrocyte volume stabilization at non-selective increase in cell membrane permeability to cations. The model included Na/K-ATPase activated by intracellular $Na^+$, $Na^+$ and $K^+$ transmembrane gradients, and took into account contribution of glycolytic metabolites and adenine nucleotides to cytoplasm osmotic pressure. We found that this model provides the best stabilization of the erythrocyte volume at non-selective increase in the permeability of the cell membrane, which can be caused by an oxidation of the membrane components or mechanical stress during circulation. The volume of the erythrocyte deviates from the optimal value by no more than 10% with a change in the non-selective permeability of the cell membrane to cations from 50 to 200% of the normal value. If only one transmembrane ion gradient is present ($Na^+$), the cell loses the ability to stabilize volume and even small changes in membrane permeability cause dramatic changes in the cell volume. Our results reveal that the presence of two oppositely directed transmembrane ion gradients is fundamentally important for robust stabilization of cellular volume in human erythrocytes.

## Introduction

The main function of erythrocytes is oxygen transportation from lungs to tissues. This function is provided due to erythrocytes ability to bind big amounts of oxygen and to circulate in the bloodstream passing through narrow tissue capillaries. The ability of erythrocytes to

Foundation (https://rscf.ru/en/) to FIA. The funders had no role in study design, data collection and analysis, decision to publish, or preparation of the manuscript.

reversibly bind oxygen is determined by a high concentration of hemoglobin in these cells (about 300 g per liter of cells [1]) and does not require energy. However, the ability of mammalian erythrocytes to circulate in the bloodstream depends on their ability to pass through narrow tissue capillaries which cross-sections frequently are smaller compared with the dimension of the erythrocytes [2–4]. The ability of erythrocytes to pass through narrow tissue capillaries is energy-dependent.

Erythrocytes of most mammalian species, including humans, have discoid shape but passing through the narrow capillaries they easily deform [2]. Erythrocytes with reduced deformability are removed from the bloodstream mainly in the spleen [4–7]. Thus, high deformability is the main criterion which determines mammalian erythrocytes usefulness and viability in an organism. The high deformability of most mammalian erythrocytes is due to elastic, but practically inextensible plasma membrane and the discoid shape of these cells [4, 8]. One should note, however, that many species in the order *Artiodactyla* (even-toed ungulates) have ellipsoid erythrocytes (family *Camelidae*), tiny spherical erythrocytes (mouse deer (family *Tragulidae*)), and other oddly shaped erythrocytes (family *Cervidae*) [9–14]. Conditions for effective circulation of such non-discoid erythrocytes was not well studied to our knowledge and, probably, high deformability is not so crucial for circulation of these cells [14].

Normal human erythrocytes have the shape of a biconcave disc with a diameter of 7–8 microns and a thickness of about 2 microns [15]. The discoid shape of an erythrocyte means that its cellular volume is significantly smaller than the volume of a sphere with a surface area equal to the surface area of its cell membrane. The volume of normal human erythrocytes is maintained within 55–60% of the maximum volume that a sphere with the same surface area as an erythrocyte has [16]. In other words, normally an erythrocyte has an excess surface relative to its volume. With an increase in cell volume by 1.7–1.8 times compared to normal, the erythrocyte takes the form of a sphere and, as a result, loses the ability to deform [3, 4, 16]. Since the erythrocyte membrane is inextensible, a further increase in cell volume leads to cell membrane rupture and cell lysis [4]. On the other hand, a decrease in the volume of the erythrocyte leads to an increase in the concentration of hemoglobin in the cytoplasm. As a result, the viscosity of the cytoplasm increases, and such erythrocyte becomes rigid and also loses the ability to deform and to pass through narrow tissue capillaries [3, 4, 16, 17]. Hard disk-shaped erythrocytes are unacceptable for a circulation as well as spherical cells. Therefore, the circulating erythrocytes should maintain an optimal ratio of the cell surface area to its volume. In the blood of a normal healthy donor, the cellular volume and surface area of circulating erythrocytes may vary more than two times, while the deviations of the ratio of surface area to volume for individual erythrocytes lies within ± 5% of the average value [7, 18–20]. In fact, this value is stabilized within the experimental error in all circulating erythrocytes. Since cells have a number of cell volume regulating systems [21], it is reasonable to assume that the erythrocyte stabilizes its volume at a given cell membrane area so as to obtain an optimal ratio of surface area to volume.

Human erythrocyte volume depends on osmotic pressure. The erythrocyte membrane permeability to water is very high [22], and the intracellular concentration of proteins and metabolites that do not penetrate through the cell membrane is significantly higher than in blood plasma. Rough estimates show that the total difference in the concentration of osmotically active components that do not penetrate the cell membrane in erythrocytes, compared with blood plasma, is about 50 mM (50 mOsm) [23, 24]. This should lead to increased osmotic pressure inside the cells. Animal cells do not try to resist osmosis. They equalize the osmotic pressure on both sides of the cell membrane, because the cell membrane breaks easily when stretched and cannot hold a pressure exceeding 2 kPa (~1 mOsm) [25]. To compensate for the osmotic pressure caused by macromolecules and metabolites, the cell could reduce the

intracellular concentration of some other substances, of which there are quite a lot both inside and outside the cell. In most cells, $Na^+$ is used as such a substance. And it would be quite natural to have a pump that removes only $Na^+$ from the cell to decrease an intracellular sodium concentration and to compensate a passive sodium transport to cytoplasm from the medium. To equalize the osmotic pressure between the cell and the medium it's sufficient to reduce sodium concentration in the cell by about 50 mM compared to the medium. In reality, the concentration of $Na^+$ in the cell is reduced to a greater extend while at the same time the cell performs seemingly meaningless work, pumping $K^+$ into the cell in almost the same amount. Here, using a mathematical model of erythrocyte volume regulation we demonstrate the importance of the existence of two oppositely directed ion gradients between the cell and the medium.

Thus, the volume of an erythrocyte is a dynamic variable and can change easily with a change in the distribution of ions between the cell and the medium. To maintain its volume, the erythrocyte must maintain ion homeostasis. In human erythrocytes, the necessary distribution of ions between the cytoplasm and the external medium is created by an ion pump–transport Na/K-ATPase, which transfers $K^+$ into the cell, and $Na^+$ from the cell to the medium in a ratio of 2:3 [26, 27], thereby reducing the total content of monovalent cations in the cell compared to the medium. As a result, the osmotic pressure outside and inside the cell is equalized. It should be noted that in the stationary state, the active fluxes of $Na^+$ and $K^+$ through the cell membrane due to the operation of the transport Na/K-ATPase should be equal to the passive transmembrane leakage of these ions (Fig 1).

A number of transport systems capable of normalizing the volume of cells placed in a hypotonic or hypertonic environment have been described in the literature [21]. In human erythrocytes such systems include K-Cl cotransport, Na-K-Cl cotransport, and Gardos channels [28]. However, the composition of the extracellular medium in the body is well stabilized and a change in its osmotic activity is rather an exception than a common situation. That explains why an erythrocyte is not protected against variations in osmolarity of an external medium and behaves in vitro as an ideal osmometer. While erythrocyte biochemical parameters, such as activities of glycolytic enzymes, Na/K-ATPase activity, ATP concentration, etc., slowly change during the cell aging in circulation, it looks like it does not affect dramatically the cell volume [16]. The question arises, what else can affect the volume of circulating erythrocytes, or from what influences (disorders / damage) should these cells be primarily protected by the volume stabilization systems available in them?

One of the causes for a significant change in the erythrocyte volume in the body may be a change of the permeability of the cell membrane for cations. During circulation, the erythrocyte membrane is exposed to high oxygen concentrations and a significant mechanical stress, that can both lead to a non-selective increase in its permeability to ions, that is, to the same increase in permeability for both $Na^+$ and $K^+$ [29–36]. Experimental and theoretical studies demonstrated that this can lead to an increase in cell volume and to the lysis of erythrocytes [23, 32, 33, 37].

Despite the presence of a number of systems capable of regulating erythrocyte volume [21], the possibility of participating in the stabilization of cell volume at a non-selective increase in the permeability of the cell membrane has not been demonstrated for any of them. Moreover, possible mechanisms of erythrocyte volume regulation are discussed in the literature mainly at the descriptive level. Earlier, using mathematical modeling, we showed that the transport Na/K-ATPase can provide stabilization of the erythrocytes cell volume at a non-selective increase in the permeability of the cell membrane for cations [23, 38]. However, that mathematical models did not take into account the contribution of glycolysis metabolites and adenine nucleotides to the osmotic pressure of the cytoplasm. Here, using the updated model, which

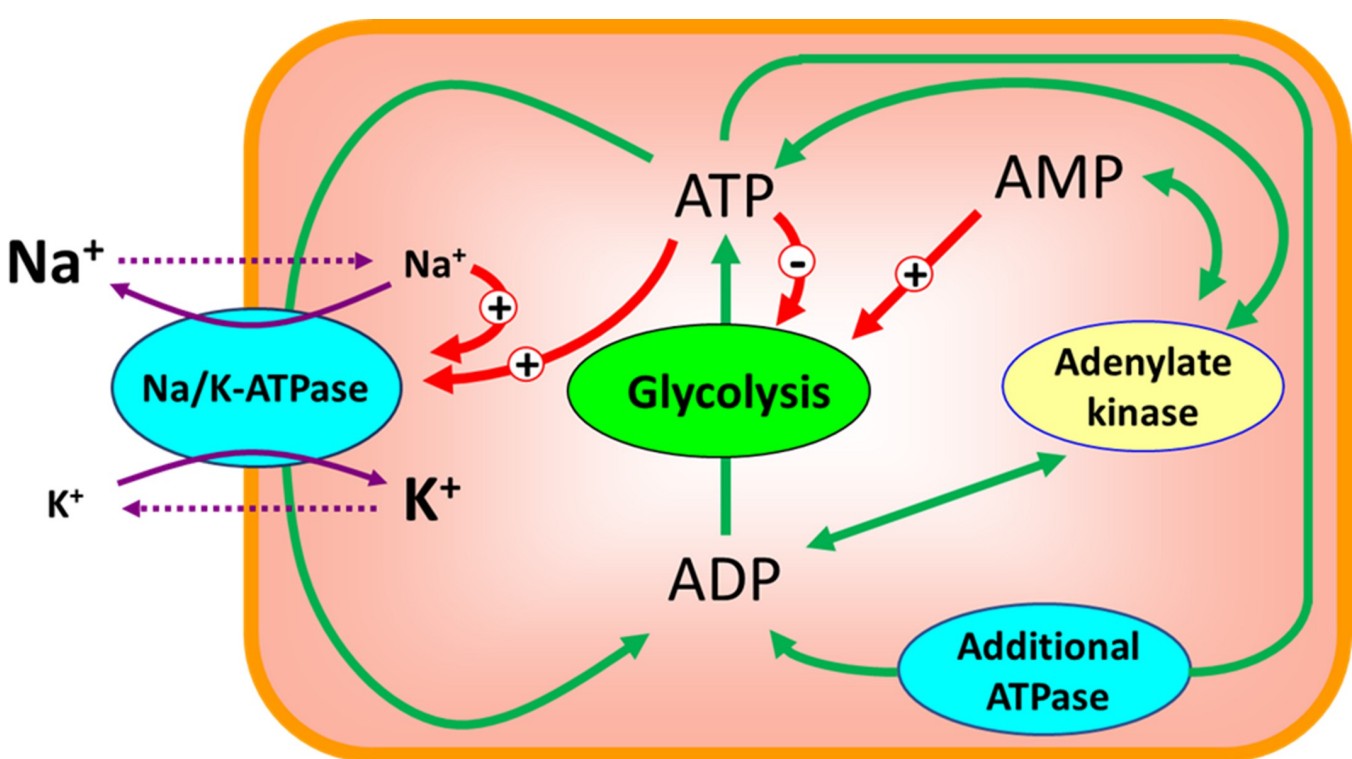

**Fig 1. Interaction of transport Na/K-ATPase and glycolysis in human erythrocytes.** Solid and dotted purple arrows show active and passive ion fluxes through the cell membrane, respectively. Ion symbols size inside and outside the cell is proportional to the ion concentration. The red arrows show activation (+) and inhibition (-) of different metabolic processes by ions and adenylates. Green arrows show the interconversions between ATP, ADP and AMP. The additional ATPase represents the ATP consuming processes in the cell other than the active transmembrane $Na^+$ and $K^+$ transport.

accounts for the contribution of glycolysis metabolites and adenine nucleotides to the osmotic pressure of the cytoplasm, we shown that the transport Na/K-ATPase provides the best stabilization of the erythrocyte volume at a non-selective increase in the permeability of the cell membrane. Moreover, we demonstrate that the presence of two oppositely directed transmembrane ion gradients ($Na^+$ and $K^+$) is crucial for the sensitivity of the cell to changes in plasma membrane permeability to cations.

## Results

To study volume regulation in human erythrocytes, we constructed a mathematical model of the metabolic and transport pathways shown in Fig 1. As expected, the model shows that non-selective increase in the cell membrane permeability leads to an increase in cell volume due to an increase in the intracellular concentration of $Na^+$ (Fig 2A and 2B). The increase in intracellular [$Na^+$] is compensated, in part, by a decrease in intracellular [$K^+$]. Also, the increase in the intracellular sodium concentration leads to an activation of the transport Na/K-ATPase (Fig 3A), which, in turn, leads to a compensation of the increased passive transmembrane fluxes of $Na^+$ and $K^+$ caused by an increase in the permeability of the cell membrane. As a result, with a twofold increase in membrane permeability, the volume of the erythrocyte increases by only 10% compared to the initial value (Fig 2A). When cell membrane permeability increases by 5 times, the volume of the erythrocyte reaches the maximum value corresponding to the spherical shape of the cell (Fig 2A), at which it completely loses the ability to deform and, consequently, to circulate in the bloodstream. Any further increase in erythrocyte volume would

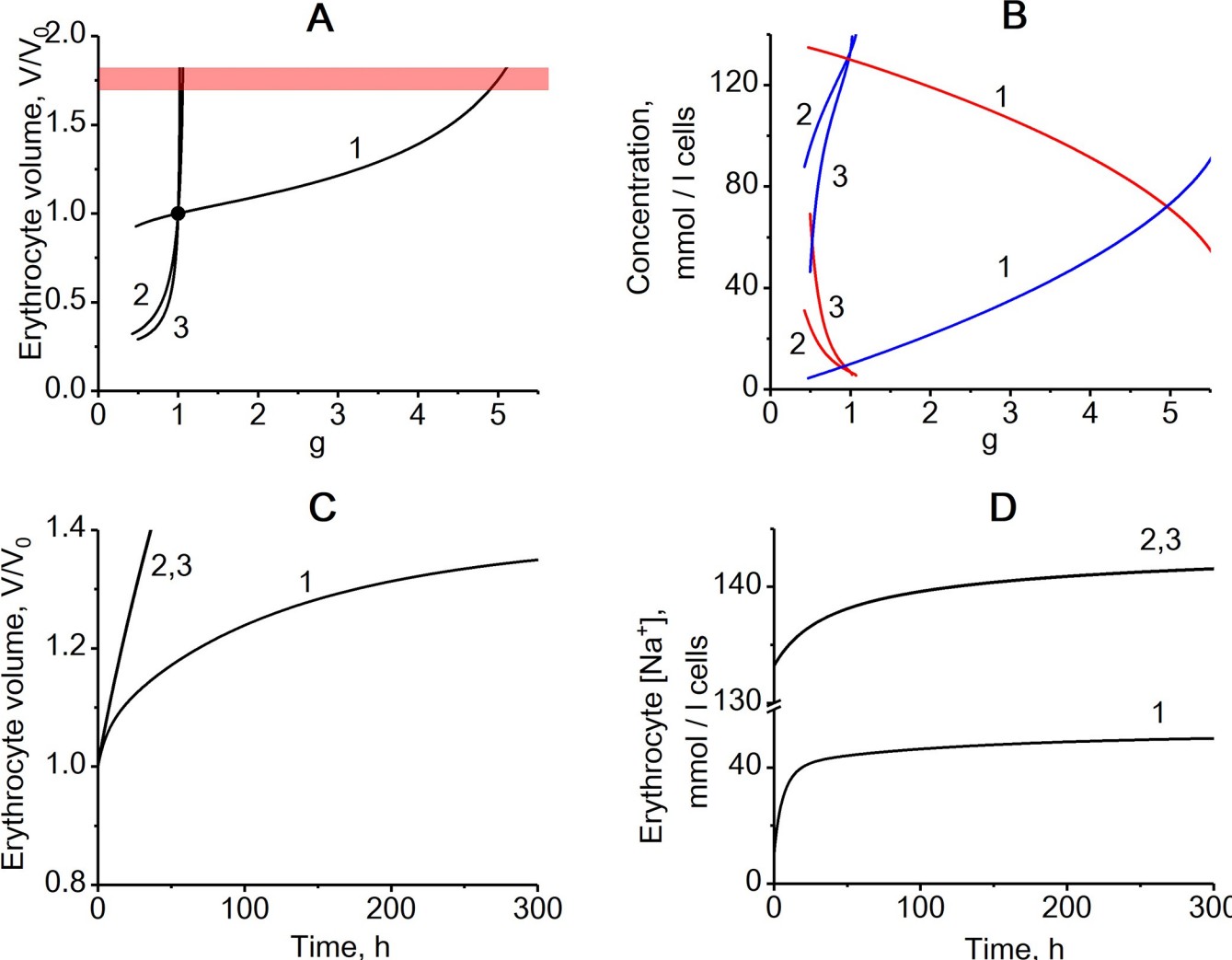

**Fig 2. The effect of non-selective permeability of the cell membrane for cations (g) on the erythrocyte volume and on intracellular $Na^+$ and $K^+$ concentrations in three different models.** The dependence of the relative stationary volume of the erythrocyte (A) and stationary intracellular $Na^+$ (blue lines) and $K^+$ (red lines) concentrations (B) on the relative non-selective permeability of the cell membrane for cations ($g = \frac{G_{Na}}{G_{Na0}} \approx \frac{G_K}{G_{K0}}$). Kinetics of changes in erythrocyte volume (C) and intracellular $Na^+$ concentration (D) after an instant 4-fold increase in the non-selective permeability of the cell membrane for cations. The kinetics of the $K^+$ concentration is the same as for $Na^+$, but changes occur in the direction of decreasing concentration. The black circle in the panel A indicates physiologically normal state of erythrocyte. The numbers on the curves correspond to the model versions: 1 –the version of the model with actively maintained transmembrane $Na^+$ and $K^+$ gradients and transport Na/K-ATPase activated by intracellular $Na^+$; 2 –version with actively maintained transmembrane gradient only for $Na^+$ and transport Na-ATPase activated by intracellular $Na^+$; 3 –version with actively maintained transmembrane gradient only for $Na^+$ and transport Na-ATPase independent of intracellular $Na^+$. The red stripe in the panel A marks the area of maximal erythrocyte volume ($V/V_0 = 1.7$–$1.8$) at which the cell takes a spherical shape.

cause its disruption. These results are similar to data obtained using models that do not take into account the contribution of glycolysis metabolites and adenylates to the osmotic pressure of the cytoplasm [23, 38], suggesting that these metabolites do not significantly contribute to erythrocytes volume stabilization under these conditions.

Now let us consider a model which includes only one active transmembrane ion gradient ($Na^+$) and an ion pump which transports only $Na^+$ out of the cell (Na-ATPase). We assume that the rate of this ATPase is proportional to the concentrations of $Na^+$ and ATP (Eq 21). Also, we assume a passive distribution of $K^+$ between cytoplasm and blood plasma in

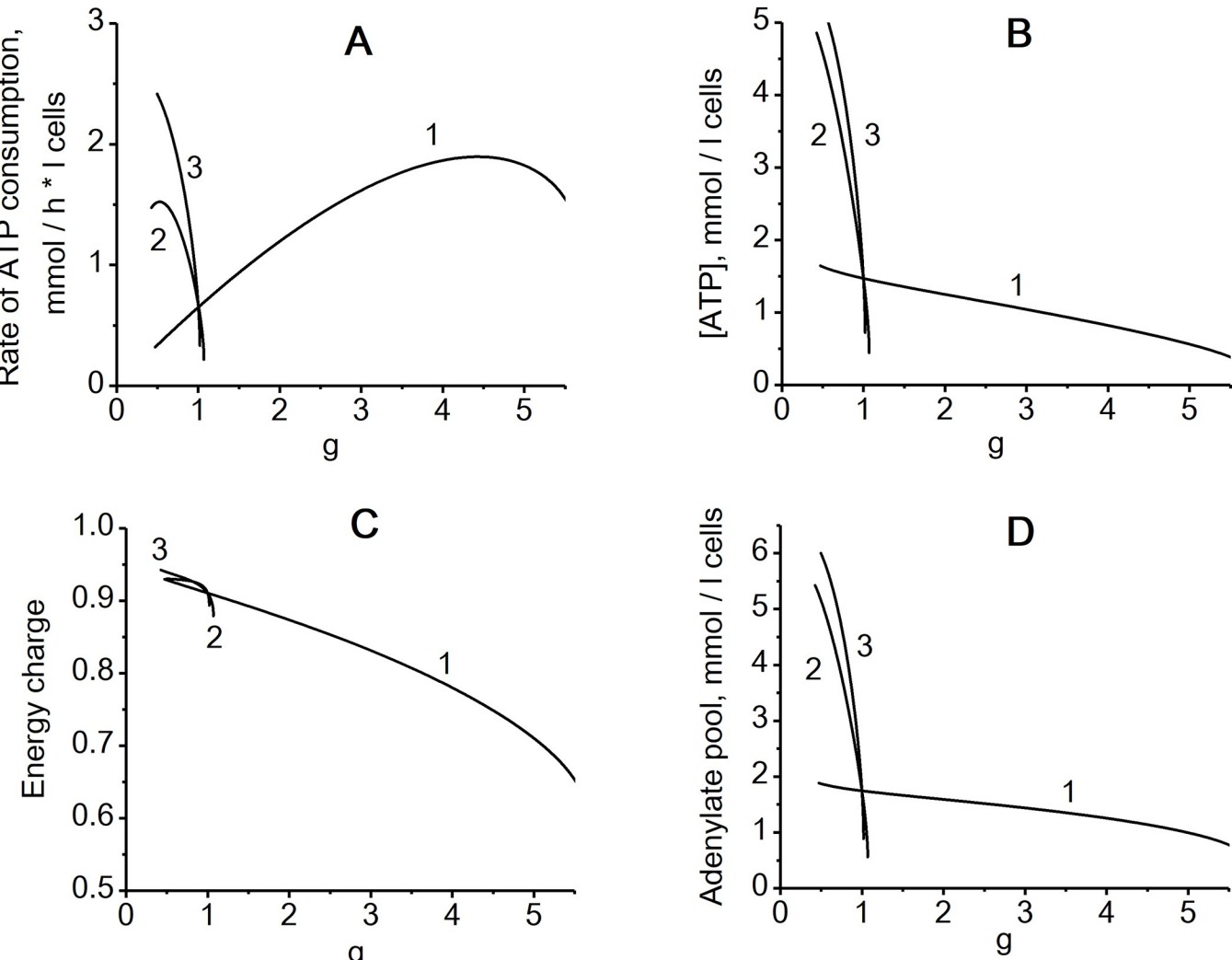

**Fig 3. The effect of non-selective permeability of the cell membrane for cations (g) on erythrocyte energy metabolism in different models.** (A)—The steady-state rate of ATP consumption by ion pumps; (B)—ATP concentration; (C)–Energy charge (([ATP] + 0.5[ADP])/([ATP] + [ADP] + [AMP])); (D)– Adenylate pool ([ATP] + [ADP] + [AMP]). The numbers on the curves indicate model versions as in Fig 2.

accordance with the transmembrane potential. In this case, steady-state intracellular $[K^+]$ is low and close to its extracellular concentration (Fig 2B). While in this model a steady-state cell volume can also be established, there is no stabilization of this volume following non-selective changes in permeability of the cell membrane for cations (Fig 2A). Even small variations of cell membrane permeability cause dramatic changes in the cell volume and intracellular ion concentrations (Fig 2A and 2B). Such lack of volume stabilization in this model system can be explained by a substantially smaller dynamic range of intracellular sodium concentration increases after a change in membrane permeability. The difference in the concentration of osmotically active components between the erythrocyte and the medium is about 50 mM, that is, about 17% of the total concentration of osmotically active components in the cell [23, 24] (Table 1). Thus, in the case of a single active transmembrane ion gradient, this gradient should be relatively small. With extracellular sodium concentration of 150 mM its intracellular concentration should be just 50 mM lower. Such a small gradient is not able to provide a significant increase in the concentration of $Na^+$ in the cell that is necessary for the effective activation

**Table 1. Model variables and parameters characterizing the normal physiological state of a human erythrocyte.**

| Variable or parameter | Normal physiological value | Comments | References |
|---|---|---|---|
| $[Cl^-]$ | 110 mmol/L cells | variable, the value was calculated from the model | |
| $[Cl^-]_{ext}$ | 150 mM | parameter | [23] |
| $E_m$ | -8,4 mV | variable | [49] |
| $G_{K0}$ | $1.24 \cdot 10^{-2}$ 1/h | parameter, the value was calculated from the model | |
| $G_{Na0}$ | $1.22 \cdot 10^{-2}$ 1/h | parameter, the value was calculated from the model | |
| $[K^+]$ | 130 mmol/L cells | variable | [23] |
| $[K^+]_{ext}$ | 5 mM | parameter | [23] |
| $[Na^+]$ | 10 mmol/L cells | variable | [23] |
| $[Na^+]_{ext}$ | 145 mM | parameter | [23] |
| T | 310˚K | parameter | |
| V | 1 L | variable | |
| W | 49 mmol | parameter, the value was calculated from the model | |
| $Z_{ADP}$ | -2 | parameter, the values for adenylates, G6P, and F6P charge were taken from the human metabolome database http://www.hmdb.ca | |
| $Z_{AMP}$ | -2 | parameter, see above | |
| $Z_{ATP}$ | -3 | parameter, see above | |
| $Z_{G6P}$ | -2 | parameter, see above | |
| $Z_{F6P}$ | -2 | parameter, see above | |
| $Z_w$ | -0.52 | parameter, the value was calculated from the model | |
| Ω | 300 mOsm | parameter | [23] |
| [ADP] | 0.25 mmol/L cells | variable | [24] |
| [AMP] | 0.041 mmol/L cells | variable | [24] |
| [ATP] | 1.5 mmol/L cells | variable | [24] |
| [G6P] | 71 µmol/L cells | variable | [24] |
| [F6P] | 23 µmol/L cells | variable | [24] |

of the transport Na-ATPase when the cell membrane is damaged (Fig 2B). Furthermore, the rate of Na-ATPase in this model decreases with an increase in cell membrane permeability due to a decrease in ATP concentration caused by an increase in cell volume and dilution of adenylates (Fig 3). Interestingly, in the model with a single transmembrane ion gradient and a regulated transport Na-ATPase the cell volume stabilization is almost as bad as in a model with a single transmembrane ion gradient and an unregulated transport Na-ATPase which rate depends on [ATP] but is independent on sodium concentration (Figs 2 and 3).

In the case of two opposite gradients of Na$^+$ and K$^+$ (that is the case in most mammalian cells), the difference in the sum concentration of Na$^+$ and K$^+$ between the cell and the medium is the same 50 mM, but the intracellular concentration of Na$^+$ is many times less than the concentration of Na$^+$ in the medium. Under these conditions, when the cell membrane is damaged, the concentration of Na$^+$ in the cell can significantly exceed its physiologically normal levels (Fig 2B) and, consequently, cause significant activation of the transport Na/K-ATPase (Fig 3A) that provides cell volume stabilization. Thus, the presence of opposite gradients of Na$^+$ and K$^+$ between the cytoplasm and the medium allows the cell to respond effectively to a damage of the cell membrane and stabilize the cell volume by activating the transport Na/K-ATPase. It is the large transmembrane gradient of Na+ that ensures the rapid and significant increase in its concentration in the cytoplasm when the cell membrane is damaged. And this gradient is achieved due to the presence of an oppositely directed transmembrane gradient of K$^+$.

We also found that the transport Na/K-ATPase, which sets the ratio of transmembrane fluxes of $Na^+$ and $K^+$ equal to 3:2, provides the best stabilization of the erythrocyte volume at a non-selective increase in the permeability of the cell membrane, when the permeability for $Na^+$ and $K^+$ increases equally (Fig 4). The cell volume stabilization is much worse if the cell membrane permeability increases predominantly for one of the ions. For example, erythrocyte volume increases significantly if permeability of cell membrane for $Na^+$ increases at unchanged permeability for $K^+$. Conversely, erythrocyte volume decreases significantly if membrane permeability for $K^+$ increases at unchanged permeability for $Na^+$ (Fig 4). If both membrane permeabilities increase simultaneously, the cell volume change very little. Thus, the cell is best protected from a non-selective increase in the cell membrane permeability.

Interestingly, in the case of only one transmembrane ion gradient, a non-selective decrease in the permeability of the cell membrane leads to a strong decrease in cell volume (curves 2 and 3 in Fig 2A). In all models, a change in the non-selective permeability of the cell membrane leads to a more or less significant change in the cell volume (Fig 2A), leading to changes in the adenylate pool value (Fig 3D), although the amount of adenylates in the cell remains constant in the models.

The characteristic time for establishing a new steady state in the model after changing the permeability of the cell membrane is tens or even hundreds of hours due to low rates of ion fluxes through the cell membrane (Fig 2C and 2D). Moreover, in the case of two transmembrane ion gradients, the release of $K^+$ from the cells partially compensates for the entry of $Na^+$ into the cells which further slows down the rate of change in cell volume.

It should be noted that the stabilization of the erythrocyte volume only due to the activity of transport Na/K-ATPase is associated with significant changes in intracellular levels of $Na^+$ and $K^+$ (Fig 2B), that is, with a disturbance of ion homeostasis in the cell, as well as with changes in the levels of ATP and the energy charge $(([ATP] + 0.5[ADP])/([ATP] + [ADP] + [AMP]))$ of the cell (Fig 3B and 3C).

Interestingly, two transmembrane ion gradients provide stabilization of the transmembrane potential (Fig 5A) at non-selective changes in cell membrane permeability that may be important for transmembrane transport processes. In the model the transmembrane potential is determined by concentration of chlorine ions in the erythrocyte (Eq (5)) that, in turn, is mainly determined by the content of $Na^+$ and K+ in the cell. In the model with Na/K-ATPase the content of cations in the erythrocyte varies slightly with non-selective change in the membrane permeability because oppositely directed passive sodium and potassium fluxes largely compensate for each other and small changes in the intracellular sodium concentration are sufficient for the Na/K-ATPase to provide compensation for the passive cation fluxes. As a result, the erythrocyte volume is stabilized (Fig 2A) as well as intracellular concentrations of macromolecules and metabolites (that is described by the ratio $W/V$ (see Table 1)), concentration of chlorine ions and transmembrane potential (Fig 5). In the model with Na-ATPase the non-selective changes in the membrane permeability associate with significant changes in the erythrocyte volume (Fig 2A) that causes significant changes in the intracellular concentrations of macromolecules and metabolites ($W/V$) (Fig 5B). Because the intracellular macromolecules and metabolites are mostly negatively charged, change in their concentrations causes an opposite change in concentration of chlorine ions. As a result, the intracellular concentration of chlorine ions and transmembrane potential also change significantly (Fig 5).

Stabilization of the erythrocyte volume in the model with two transmembrane ion gradients is not very sensitive to variation of the model parameter values such as activity of transport Na/K-ATPase or activities of glycolytic enzymes. Fig 6 demonstrates that cell volume stabilization is maintained even at twofold increase or decrease in hexokinase (HK) or transport Na/K-ATPase activity. Actually, an increase in Na/K-ATPase activity provides more possibilities

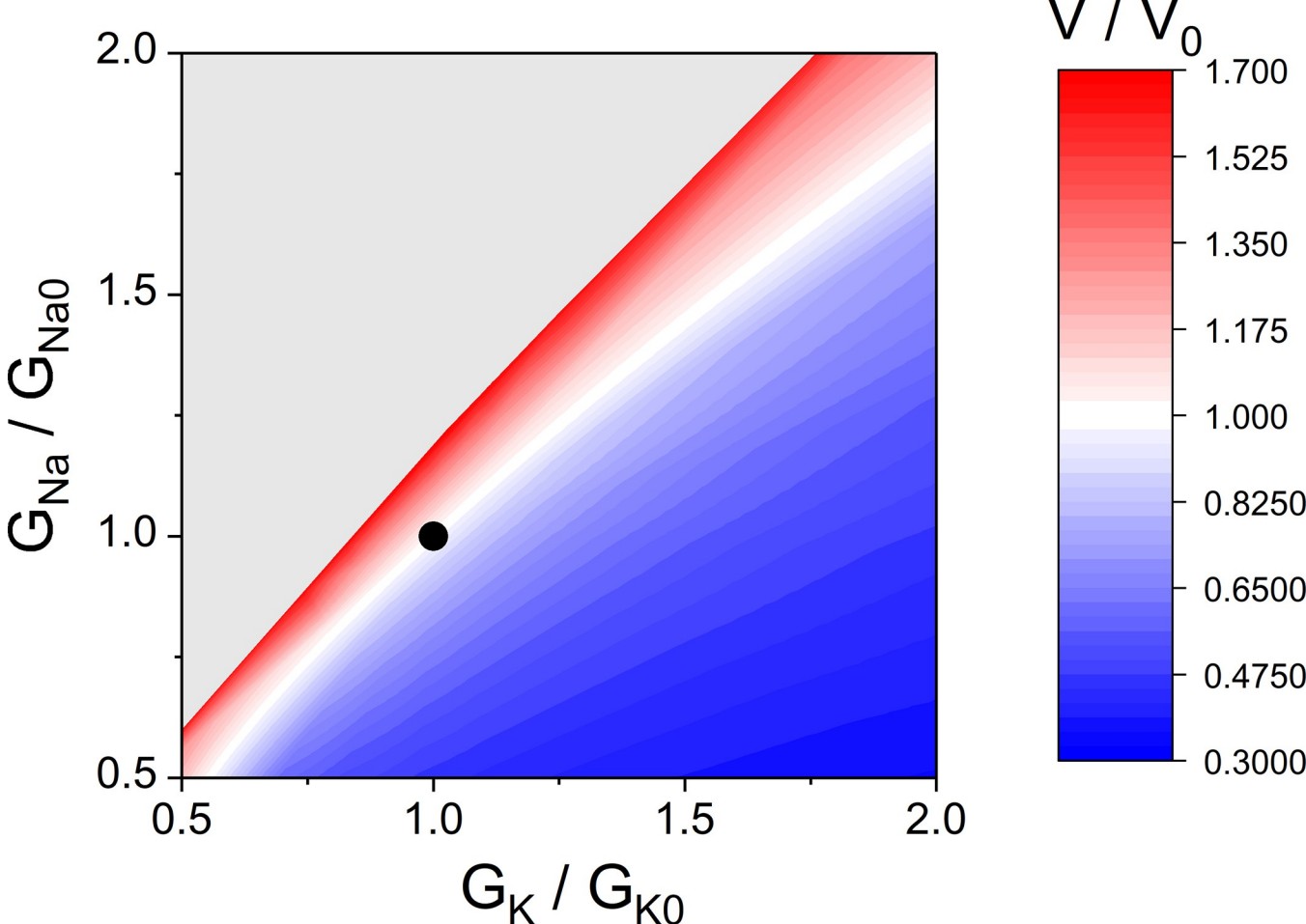

**Fig 4. The dependence of the relative steady-state erythrocyte volume ($V/V_0$) on the passive permeability of the cell membrane for $K^+$ ($G_K/G_{K0}$) and $Na^+$ ($G_{Na}/G_{Na0}$).** The model includes actively maintained transmembrane $Na^+$ and $K^+$ gradients and transport Na/K-ATPase activated by intracellular $Na^+$ (Version 1). The black circle indicates the normal physiological state of the erythrocyte.

for ion transport activation at increased cell membrane permeability and, thus, improves cell volume stabilization (Fig 6A). Volume stabilization gets worse at decreased Na/K-ATPase activity but still is pronounced. Hexokinase activity determines the maximal rate of glycolysis. An increase in hexokinase activity causes an increase in intracellular ATP levels and provides better support for activation of Na/K-ATPase and also improves erythrocyte volume stabilization (Fig 6B), while at decreased hexokinase activity the volume stabilization gets worse but still pronounced. Also, in the model with two transmembrane ion gradients and Na/K-ATPase the physiological volume of erythrocyte is almost independent on activities of Na/K-ATPase and hexokinase (Fig 6C and 6D). Indeed, variations in the Na/K-ATPase activity in the model are equivalent to variations in non-specific cell membrane permeability to cations that do not affect relative $Na^+$ and $K^+$ fluxes across cell membrane and almost do not affect ion content in cytoplasm (Fig 4). Similarly, variations in HK activity affect the maximal rate of glycolysis and activates the Na/K-ATPase due to an increase in ATP levels but do not affect ion content in cytoplasm. Contrary to that, in the case of one transmembrane ion gradient and Na-ATPase the variations in the ATPase activity as well as variations in HK activity cause asymmetric changes in $Na^+$ and $K^+$ fluxes and even physiological cell volume cannot be stabilized at variations of the model parameters (Fig 6C and 6D).

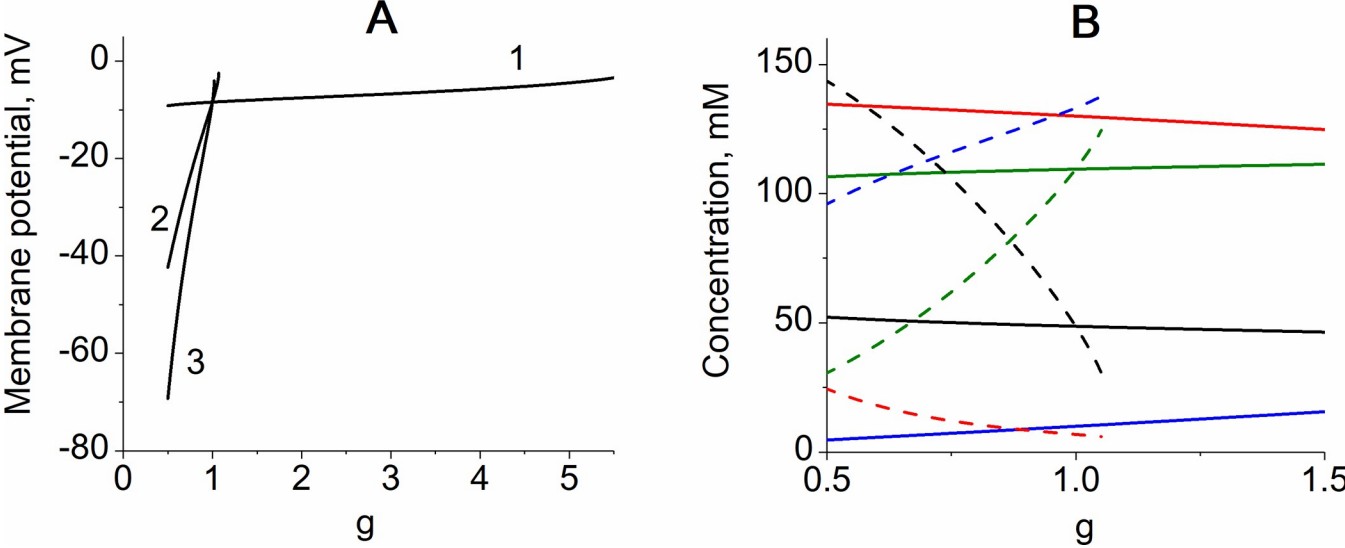

**Fig 5.** The dependence of erythrocyte transmembrane potential (A) and intracellular ion concentrations (B) on the non-selective cell membrane permeability for cations (g) in different models. (A)-The numbers on the curves indicate model versions as in Fig 2; (B)-the red, blue, green, and black lines show concentrations of $K^+$, $Na^+$, $Cl^-$, and total concentration of intracellular macromolecules and metabolites (*W/V*) respectively. Solid and dashed lines show data obtained for the model with Na/K-ATPase (model version 1), and with Na-ATPase (model version 2) respectively.

Finally, the model with transport Na/K-ATPase and two transmembrane ion gradients predicts high sensitivity of human erythrocyte volume to changes in the content of impermeable molecules inside the cell or osmolarity of the external medium change (Fig 7) that is a well-known experimental fact demonstrating that human erythrocytes are not protected against changes in osmolarity of cytoplasm or external medium.

## Discussion

Our results show that the presence of two oppositely directed active transmembrane ion gradients ($Na^+$ and $K^+$) and the transport Na/K-ATPase activated by intracellular sodium are fundamentally important for the stabilization of cellular volume in human erythrocytes. Under these conditions, the most effective stabilization of the cell volume is provided at a non-selective increase in the permeability of the cell membrane. It seems to us that a non-selective increase in the permeability of erythrocyte membranes is the most likely damage that the cell membranes endure during circulation in the bloodstream. Based on these results, a general conclusion can be done that the presence of two oppositely directed transmembrane ion gradients ($Na^+$ and $K^+$) at a low intracellular concentration of the ion prevailing in the external medium ($Na^+$) provides a greater (compared to conditions with a single gradient ($Na^+$)) sensitivity of the cell to a damage of the cell membrane and is a fundamentally necessary condition for ensuring the stabilization of cell volume and preventing cell destruction. One can assume that such cell organization arose in the early stages of evolution and later served as the basis for the emergence of cellular electrical excitability. Also, this result casts doubt on the hypothesis that the high level of potassium in cells reflects the ionic composition of the environment in which the first cellular organisms originated [39–41]. From the results presented here, it follows that in order to maintain cellular volume and preserve the integrity of cells, it would be very impractical for primary organisms to maintain an intracellular ionic composition similar to that of the external milieu.

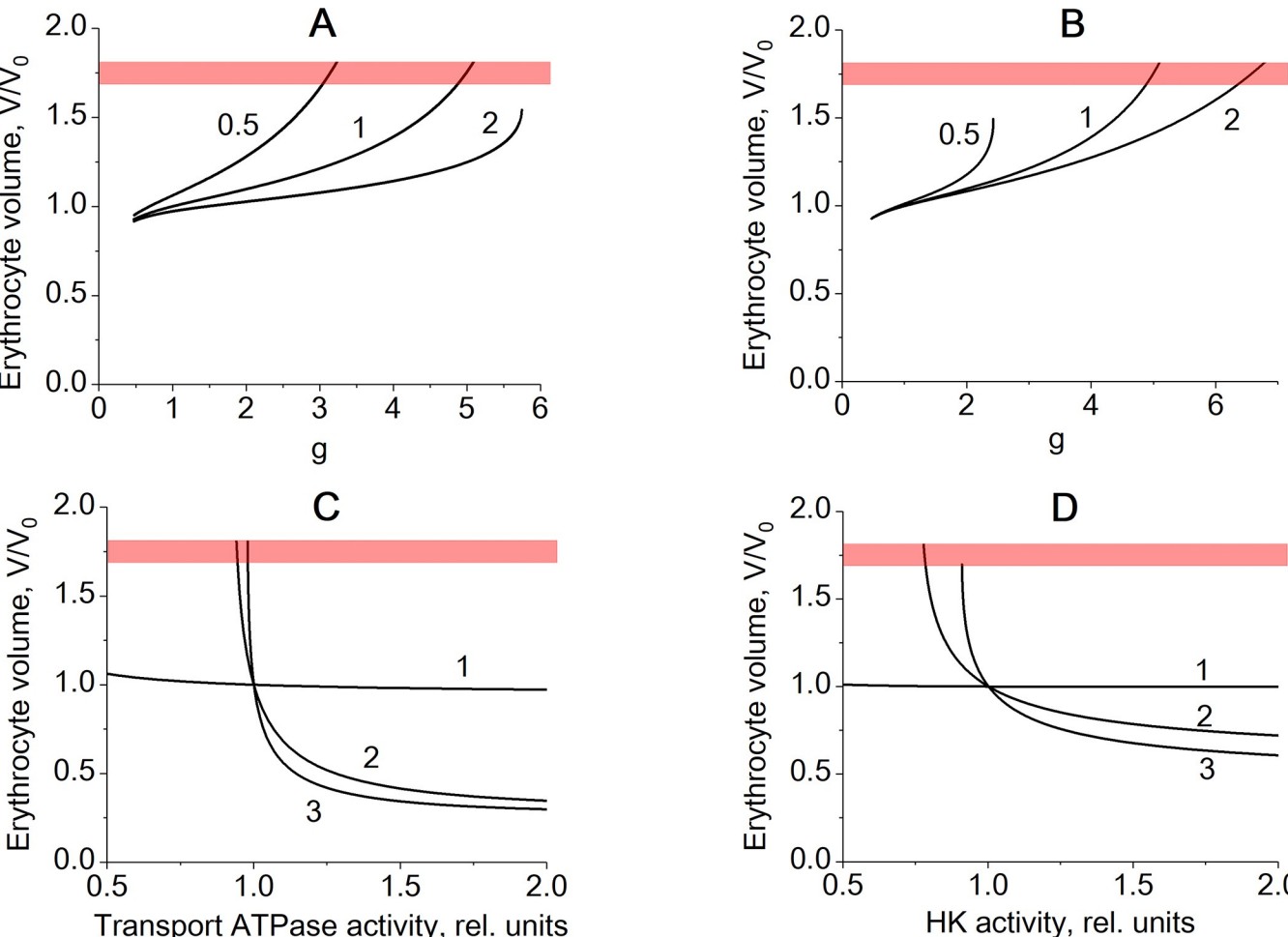

**Fig 6. The effect of variation of transport Na/K- or Na- ATPase and hexokinase (HK) activity on erythrocyte volume stabilization.** Panels (A) and (B) show the dependence of the relative erythrocyte cell volume on the non-selective cell membrane permeability for cations (g) at different values of transport Na/ K-ATPase activity (A) or HK activity (B) in the base version of the model (model version 1). The numbers on the curves indicate relative activity of the corresponding enzyme. Panel (C) shows the dependence of the physiological erythrocyte volume on the activity of the transport Na/K-ATPase (model version 1, curve 1) or transport Na-ATPase (model versions 2 and 3). Panel (D) shows the dependence of the physiological erythrocyte volume on HK activity in the models with Na/K-ATPase (curve 1) or with Na-ATPase (curves 2 and 3). The red stripes mark the area of maximal erythrocyte volume ($V/V_0$ = 1.7–1.8) at which the cell takes a spherical shape. For curve 2 (panel A) and for curve 0.5 (panel B) at high g values as well as for curve 3 (panel D) at low HK activity values the steady state in glycolysis disappears.

Nevertheless, Na/K-ATPase alone is unable to provide stabilization of the erythrocyte volume in a sufficiently wide range of changes in the permeability of the cell membrane. Mathematical simulation shows that at two-fold increase in membrane permeability compared to the normal value, the cell volume increases by only ~10% (Fig 2A). However, this already goes beyond the ±5% frame in which the volume of the erythrocyte to its surface area ratio is stabilized in the body [7, 18–20]. Similarly, our model predicts that an increase in cell membrane permeability of more than 5-fold leads to cell lysis, whereas literature data indicate that erythrocytes can remain in the bloodstream with an increase in the permeability of the cell membrane of more than 5–10 times compared to the normal value [42–45]. The results of our previous studies show that taking into account the additional ion transport system (calcium-activated potassium channels or Gardos channels) and the metabolism of adenine nucleotides in the model makes it possible to achieve almost perfect stabilization of the erythrocyte volume

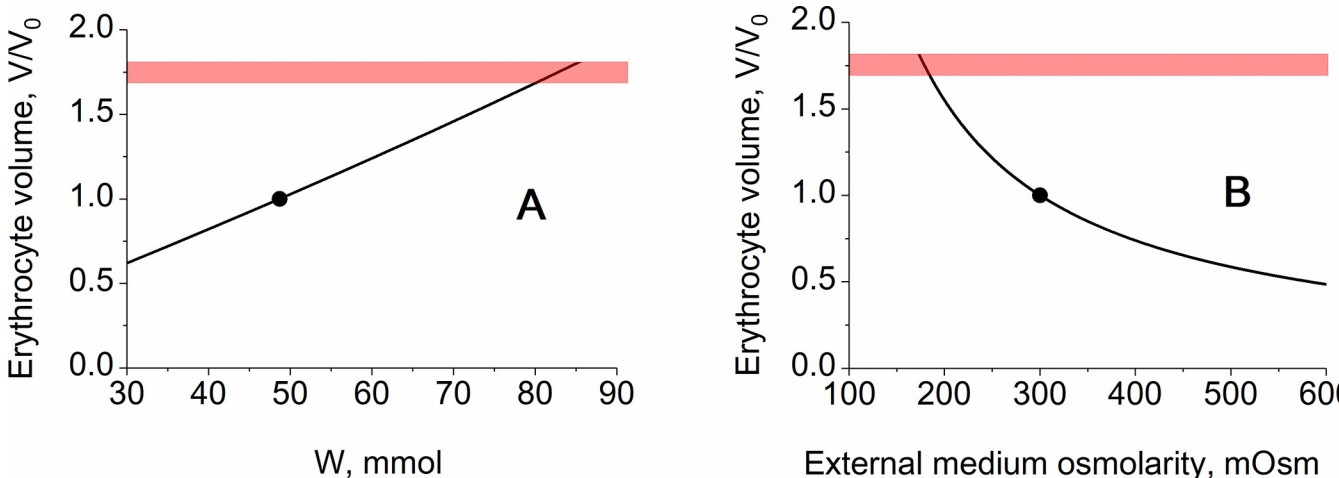

**Fig 7. The dependence of the erythrocyte steady-state volume on the content of impermeable molecules in cytoplasm (A) and on the osmolarity of the external medium (B) in the model with two ion gradients.** In panel (A) parameter $W$ (Table 1) was varied above and below its physiological value. In panel (B) parameters $[Na^+]_{ext}$, $[K^+]_{ext}$, and $[Cl^-]_{ext}$ (Table 1) were varied above and below the physiological values simultaneously at the same proportion. The black circles indicate physiologically normal state of erythrocyte. The red stripes mark the area of maximal erythrocyte volume ($V/V_0 = 1.7-1.8$) at which the cell takes a spherical shape.

with an increase in the permeability of the cell membrane up to 15 times compared to the normal value [23, 38]. In this regard, the role of Gardos channels, as well as other ion transport systems, and adenylate metabolism in stabilizing the volume of human erythrocytes should be revised using more correct models.

## Methods (Mathematical model description)

Mathematical model used in this study is a system of algebraic and ordinary differential equations that describe transmembrane ion fluxes, osmotic regulation of human erythrocyte cell volume and glycolysis.

### Ions and cell volume regulation

The description of ion balance and cell volume regulation is based on our earlier mathematical model of human erythrocyte volume regulation [23] and takes into account contribution of adenine nucleotides and glycolytic metabolites to osmotic balance. Eq (1) describes equivalence of osmolality in cytoplasm and in external medium (blood plasma) while the Eq (2) describes the cytoplasm electroneutrality:

$$\sum c_i + \frac{W}{V} = \Omega \qquad (1)$$

$$\sum z_i c_i + z_w \frac{W}{V} = 0 \qquad (2)$$

Here $c_i$ denotes concentration of each osmotically active cytoplasm component, which is described in the model as a variable. They are $Na^+$, $K^+$, and $Cl^-$, adenylates–ATP, ADP, AMP, and glycolysis intermediates–glucose-6-phosphate (G6P) and fructose-6-phosphate (F6P). $W$ denotes total amount of all other osmotically active components of human erythrocytes, including hemoglobin, enzymes, and metabolites, which kinetics is not described explicitly in the model. $Z_i$ denotes electrical charge of the corresponding osmotically active component. $Z_W$

denotes the average electrical charge of components not explicitly described in the model. $\Omega$—total concentration of osmotically active components in blood plasma. $V$–erythrocyte volume. It is more convenient to express volume in the model as per $10^{13}$ erythrocytes that is equal to one liter rather than per one erythrocyte. Thus, under normal physiological conditions $V = V_0$ = 1 L. In this case, the amounts of substances expressed in grams in the cells are numerically equal to their molar concentrations.

Eqs (3 and 4) describe kinetics of quantity of $Na^+$ and $K^+$ in erythrocytes:

$$\frac{d([Na^+]V)}{dt} = -3U_{Na/K-ATPase} + J_{Na} \tag{3}$$

$$\frac{d([K^+]V)}{dt} = 2U_{Na/K-ATPase} + J_K \tag{4}$$

Here $U_{Na/K\text{-}ATPase}$–the rate of ATP consumption by transport Na/K-ATPase, $J_{Na}$ and $J_K$ are passive $Na^+$ and $K^+$ fluxes across cell membrane, respectively. The cell membrane permeability for $Cl^-$ anions in the erythrocyte is two orders of magnitude higher than for cations [37], therefore Eq (5) describes the equilibrium distribution of chlorine between the cytoplasm and the medium in accordance with the electric potential on the membrane:

$$\frac{[Cl^-]}{[Cl^-]_{ext}} = e^{\frac{FE_m}{RT}} \tag{5}$$

Here $[Cl^-]$ and $[Cl^-]_{ext}$–intracellular and extracellular concentrations of chlorine ions, respectively, $F$–Faraday constant, $E_m$—electrical potential on the erythrocyte cell membrane, $R$–universal gas constant, $T$—absolute temperature (310 K). Equation for the rate of Na/K-ATPase and parameter value were taken from [23]:

$$U_{Na/K-ATPase} = A_{Na/K\text{-}ATPase}[Na^+][ATP] \tag{6}$$

$$A_{Na/K\text{-}ATPase} = 0.044 \text{ L}^2 \text{ / h mmol}$$

Here $A_{Na/K\text{-}ATPase}$ denotes activity of the Na/K-ATPase. Hereafter, $A_X$ denotes the activity of the enzyme X.

The passive ion fluxes across the erythrocyte cell membrane are described in Goldman approximation regarding constancy of the electric field inside the cell membrane [46]:

$$J_i = G_i \frac{r}{e^r - 1} \left([I]_{ext} - [I]e^r\right), \tag{7}$$

$$r = \frac{Z_i FE_m}{RT}$$

Here $G_i$–erythrocyte membrane permeability for ion $I$. $[I]$ and $[I]_{ext}$–concentration of ion $I$ inside and outside of the erythrocyte, correspondingly.

## Glycolysis and energy metabolism

In the model we used the simplified description of glycolysis. Only the upper part of glycolysis is explicitly described, including hexokinase, glucosephosphateisomerase, and phosphofructokinase reactions—which determine the rate of the entire glycolysis. Eqs (8 and 9) describe

kinetics of quantity of G6P and F6P molecules in the erythrocyte respectively:

$$\frac{d([G6P]V)}{dt} = U_{HK} - U_{GPI} \tag{8}$$

$$\frac{d([F6P]V)}{dt} = U_{GPI} - U_{PFK} \tag{9}$$

Here $U_{HK}$, $U_{GPI}$, and $U_{PFK}$ denote hexokinase, glucosephosphateisomerase, and phosphofructokinase reaction rates in glycolysis, respectively. We neglect the metabolic flow in the 2,3-diphosphoglycerate shunt and assume that the rate of reactions of the lower part of glycolysis (from aldolase to lactatedehydrogenase), is equal to twice the rate of the phosphofructokinase reaction. Then the rate of ATP production in glycolysis is determined by the difference between the rates of its production in phosphoglyceratekinase ($U_{PGK}$) and pyruvate kinase ($U_{PK}$) reactions and consumption in hexokinase and phosphofructokinase reactions:

$$-U_{HK} - U_{PFK} + U_{PGM} + U_{PK} = -U_{HK} + 3U_{PFK} \tag{10}$$

Total amount of $ATP + ADP + AMP$ in the cells was assumed to be constant. We also assumed that rapid adenylate equilibrium exists in the cells all the time (Eq (11)).

$$[ATP][AMP] = [ADP]^2 \tag{11}$$

The energy balance in the cell can be described by the following equation:

$$\frac{d(qV)}{dt} = -U_{HK} + 3U_{PFK} - U_{Na/K-ATPase} - U_{ATPase} \tag{12}$$

Here $q = 2[ATP] + [ADP]$, $U_{ATPase}$–the rate of additional ATPase reaction added to the model to balance the rates of ATP production and consumption [23, 24, 47, 48].

The presence of rapid adenylatekinase equilibrium in the cell leads to the fact that of the three variables–ATP, ADP, and AMP–only two are independent. The transition to the variables $q$ and $p$ = [ATP] + [ADP] + [AMP] (adenylate pool) makes it possible to exclude the rapid adenylate kinase reaction from the equations. Concentrations of ATP, ADP, and AMP at fixed values of $q$ and $p$ can be calculated from the following system of equations:

$$q = 2[ATP] + [ADP]$$

$$p = [ATP] + [ADP] + [AMP] \tag{13}$$

$$[ATP][AMP] = [ADP]^2$$

The equations for the rates of HK, GPI, PFK, and additional ATPase reactions and parameter values were taken from [24]:

$$U_{HK} = A_{HK} \frac{[ATP]/K_{HK1}}{1 + [ATP]/K_{HK1} + [G6P]/K_{HK2}} \tag{14}$$

Here $A_{HK}$ = 12 mmol / h, $K_{HK1}$ = 1 mM, $K_{HK2}$ = 5.5 μM.

$$U_{GPI} = A_{GPI} \frac{([G6P] - [F6P]K_{GPI1})/K_{GPI2}}{1 + \frac{[G6P]}{K_{GPI2}} + \frac{[F6P]}{K_{GPI3}}} \tag{15}$$

Here $A_{GPI}$ = 360 mmol / h, $K_{GPI1}$ = 3 mM, $K_{GPI2}$ = 0.3 mM, $K_{GPI3}$ = 0.2 mM

$$U_{PFK} = A_{PFK} \, 1.1 \frac{[F6P]}{[F6P] + K_{PFK1}} \cdot \frac{[ATP]}{[ATP] + K_{PFK2}} \cdot \frac{\frac{1}{1+[AMP]/K_{PFK3}} + 2\frac{[AMP]}{[AMP]+K_{PFK3}}}{1 + 10^8 \left(\frac{(1+[ATP]/K_{PFK4})}{(1+[AMP]/K_{PFK3})(1+[F6P]/K_{PFK5})}\right)^4} \quad (16)$$

Here $A_{PFK}$ = 380 mmol / h, $K_{PFK1}$ = 0.1 mM, $K_{PFK2}$ = 2 mM, $K_{PFK3}$ = 0.01 mM, $K_{PFK4}$ = 0.195 mM, $K_{PFK5}$ = 0.37 μM.

$$U_{ATPase} = A_{ATPase} \frac{[ATP]}{[ATP] + K_{ATPase}} \quad (17)$$

Here $A_{ATPase}$ = 2.7 mmol / h, $K_{ATPase}$ = 1 mM

## Calculation of the model parameter values

To finalize the model, it is necessary to calculate some of its parameters–these are the normal physiological values of the intracellular concentration of chlorine ions, the permeability of the erythrocyte membrane for cations, the $W$ value and the average charge of osmotically active components that do not penetrate the erythrocyte membrane ($Z_W$), such as hemoglobin, enzymes and metabolites. Concentration of chlorine ions in the erythrocyte can be calculated from Eq (5), taking extracellular concentration of chlorine of 150 mM, the membrane potential of -8.4 mV and the temperature of 310 K [23, 49] (Table 1). Then, values of $W$ and $Z_W$ are calculated from Eqs (1) and (2), using the physiological values of the variables from Table 1. According to these calculations $W$ = 49 mmol and $Z_W$ = 0.52.

## Reduction of a mathematical model to a system of ordinary differential equations

Initially the mathematical model includes five differential (3, 4, 8, 9, 12) and three algebraic (1, 2, 5) equations. If one chooses the amounts of substances in the erythrocyte as the model variables:

$$Q_K = [K^+]V, Q_{Na} = [Na^+]V, Q_{Cl} = [Cl^-]V,$$

$$Q_{G6P} = [G6P]V, Q_{F6P} = [F6P]V, Q_q = qV, Q_p = pV \quad (18)$$

and express the variables $V$, $Q_{Cl}$, and $E_m$ through the remaining variables using algebraic Eqs (1, 2 and 5), then the model is transformed into a system of five ordinary differential equations, similar to (3, 4, 8, 9, 12), in respect to variables $Q_K$, $Q_{Na}$, $Q_{G6P}$, $Q_{F6P}$, $Q_q$.

Investigation of the kinetic behavior of the model was performed using the CVODE software [50], and investigation of the dependence of the steady state of the model on the parameters was performed using the AUTO software [51].

## Description of the cell membrane damage in the model

The main task of this work was to study the effect of non-specific damage of the cell membrane on erythrocyte volume. As it is noted in the introduction, we assume that such damage leads to a non-selective increase in the permeability of the cell membrane for all cations as:

$$G_{Na} = G_{Na0} + \Delta, \ G_K = G_{K0} + \Delta \quad (19)$$

Here $G_{Na0}$ and $G_{K0}$ are physiological values of cell membrane permeability for Na$^+$ and K$^+$ respectively and $\Delta$ denotes a value of an increase in the membrane permeability for cations.

Generally speaking, both an increase and a decrease in the cell membrane permeability can be described in this way. For clarity of presentation, we also introduced the parameter *g* corresponding to the relative permeability of the membrane for cations:

$$g = \frac{G_{Na}}{G_{Na0}} \approx \frac{G_{KP}}{G_{KP0}} \tag{20}$$

The validity of Eq (20) follows from the fact that $G_{Na0} \approx G_{K0}$ (Table 1).

## The mathematical model versions

*Version 1.* The basic version of the model describing an erythrocyte with transmembrane transport Na/K-ATPase activated by Na$^+$, glycolysis, and constant adenylate content. The model consists of Eqs (3, 4, 8, 9 and 12) modified according to the Eq (18).

*Version 2.* The modified model where Na/K-ATPase was replaced with the transmembrane transport Na-ATPase (a pump that transports only Na$^+$ from the cell to external medium) which rate is proportional to intracellular concentrations of ATP and Na$^+$. The rate of this pump is described by the following equation:

$$U_{Na-ATPase2} = A_{Na-ATPase2}[ATP][Na^+], \tag{21}$$

where $A_{Na-ATPase2}$ = 0.0035 L$^2$ / h

The value of this parameter was chosen in such a way that this ATPase compensates for the passive flow of Na$^+$ into the cell under physiological conditions.

In this version of the model the Eq (4) does not contain a term describing active potassium transmembrane transport. Thus, the rate of change in intracellular potassium concentration is determined only by a passive potassium flux across the cell membrane according to the Eq (7). Under these conditions the steady-state potassium concentration in the cells was calculated under assumption of zero passive potassium flux across the cell membrane according to the Eq (7):

$$[K] = [K]_{ext}\, e^{-\frac{FE_m}{RT}} \tag{22}$$

And the steady-state sodium concentration was calculated as follows:

$$[Na] = 140 - [K] \tag{23}$$

Here 140 (mmol/L cells) is a sum of Na$^+$ and K$^+$ concentrations in erythrocytes containing Na/K-ATPase under normal conditions. This condition guarantees that the erythrocyte will have the same physiological volume and transmembrane potential in all model versions.

*Version 3.* A model where the Na/K-ATPase was replaced with the transmembrane transport Na-ATPase (as in Version 2) which rate depends on [ATP] but does not depend on sodium concentration. The rate of this pump is described as:

$$U_{Na-ATPase3} = A_{Na-ATPase3}[ATP] \tag{24}$$

where $A_{Na-ATPase3}$ = 0.46 L / h

The value of this parameter was chosen in such a way that this ATPase compensates for the passive flow of Na$^+$ into the cell under physiological conditions. In this version of the model the intracellular potassium and sodium concentrations were calculated using the Eqs (23) and (24) the same as in Version 2.

## Author Contributions

**Conceptualization:** F. I. Ataullakhanov.

**Data curation:** F. I. Ataullakhanov, M. V. Martinov, V. M. Vitvitsky.

**Formal analysis:** F. I. Ataullakhanov, M. V. Martinov, Qiang Shi, V. M. Vitvitsky.

**Investigation:** F. I. Ataullakhanov, M. V. Martinov, V. M. Vitvitsky.

**Methodology:** M. V. Martinov.

**Project administration:** F. I. Ataullakhanov.

**Software:** M. V. Martinov.

**Supervision:** F. I. Ataullakhanov.

**Writing – original draft:** V. M. Vitvitsky.

**Writing – review & editing:** Qiang Shi.

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
