## [Decision Letter · Decision Letter 0]

13 Sep 2022

PONE-D-22-20774Significance of two transmembrane ion gradients for human erythrocyte volume stabilization.PLOS ONE

Dear Dr. Vitvitsky

Thank you for submitting your manuscript to PLOS ONE. After careful consideration, we feel that it has merit but does not fully meet PLOS ONE’s publication criteria as it currently stands. Therefore, we invite you to submit a revised version of the manuscript that addresses the points raised during the review process.

<h3>**Your manuscript entitled “Significance of two transmembrane ion gradients for human erythrocyte volume stabilization."has been reviewed by two Reviewers. The Reviewers were of the opinion that the manuscript contains important information that is likely to be of interest to other investigators. However, they also identified a number of concerns that require further attention, as indicated in the reviews appended below**.

**Please modify the manuscript to address all of the reviewers' concerns that includes a point-by-point response to all of the issues raised in the reviews and submit the revised version via the PLOS ONE Electronic Submission site. **</h3>

We look forward to receiving your revised manuscript.

Kind regards,

Pramod K. Yadav, Ph. D.

Academic Editor

PLOS ONE

Journal Requirements:

2. Please note that PLOS ONE has specific guidelines on code sharing for submissions in which author-generated code underpins the findings in the manuscript. In these cases, all author-generated code must be made available without restrictions upon publication of the work. Please review our guidelines at https://journals.plos.org/plosone/s/materials-and-software-sharing#loc-sharing-code and ensure that your code is shared in a way that follows best practice and facilitates reproducibility and reuse. New software must comply with the Open Source Definition.

3. Please ensure that you have specified (1) whether consent was informed and (2) what type you obtained (for instance, written or verbal, and if verbal, how it was documented and witnessed). If your study included minors, state whether you obtained consent from parents or guardians. If the need for consent was waived by the ethics committee, please include this information.

"This work was supported in part by grant No 21-45-00012 from the Russian Science Foundation (https://rscf.ru/en/) to FIA."

"This work was supported in part by grant from the Russian Science Foundation № 21-45-00012 (F.I. Ataullakhanov)."

"This work was supported in part by grant No 21-45-00012 from the Russian Science Foundation (https://rscf.ru/en/) to FIA."

7. PLOS requires an ORCID iD for the corresponding author in Editorial Manager on papers submitted after December 6th, 2016. Please ensure that you have an ORCID iD and that it is validated in Editorial Manager. To do this, go to ‘Update my Information’ (in the upper left-hand corner of the main menu), and click on the Fetch/Validate link next to the ORCID field. This will take you to the ORCID site and allow you to create a new iD or authenticate a pre-existing iD in Editorial Manager. Please see the following video for instructions on linking an ORCID iD to your Editorial Manager account: https://www.youtube.com/watch?v=_xcclfuvtxQ

Reviewers' comments:

Reviewer's Responses to Questions

**Comments to the Author**

1. Is the manuscript technically sound, and do the data support the conclusions?

Reviewer #1: Yes

Reviewer #2: Yes

2. Has the statistical analysis been performed appropriately and rigorously? 

Reviewer #1: N/A

Reviewer #2: Yes

3. Have the authors made all data underlying the findings in their manuscript fully available?

Reviewer #1: Yes

Reviewer #2: Yes

4. Is the manuscript presented in an intelligible fashion and written in standard English?

Reviewer #1: Yes

Reviewer #2: Yes

5. Review Comments to the Author

Reviewer #1: The paper by Ataullakhanov et al entitled “Significance of two transmembrane ion gradients for human erythrocyte volume stabilization” is a mathematical modeling study of the role of ionic gradients in stabilization of erythrocyte volume. Maintenance of an optimal surface-to-volume ratio is vital for the physiological function of erythrocytes as it allows them to deform easily inside the capillaries that are several-fold smaller in diameter than the cell if it were a sphere. This manuscript focuses on the benefits of having two (Na+/K+) transmembrane gradients instead of just one (Na+) and demonstrates that larger dynamic range of Na+ concentrations provided by the activity of Na+/K+-ATPase is critical for RBC volume stabilization under several-fold increases in non-specific membrane permeability to ions. The study is very interesting and has applications for other cells, such as neurons where large dynamic range of transmembrane ionic gradients is similarly physiologically vital. I have several suggestions that in my opinion will improve the manuscript and make it more accessible for wider audiences.

Major:

1. While the model describes transmembrane osmotic gradients, electrical component of ionic gradients is not discussed in the main text. The model does incorporate an algebraic equation for electroneutrality inside the cell and the dependence of passive ion fluxes on membrane potential, but it appears that erythrocyte membrane potential is a constant rather than the variable, which is a significant limitation of this study that the authors should discuss.

2. As the focus of the manuscript is on S/V stabilization, I am curious to see how robust the model is to account for natural cell-to-cell variability. For example, how well is S/V maintained when Na+/K+-ATPase surface density is changed or when intracellular “passive” osmotic pressure is varied?

Minor:

1. I don’t see much added value in the 3rd model (Na+/K+-ATPase without Na+-mediated activation). The authors should consider moving these data to supplementary materials or removing them from the manuscript.

2. Can the authors speculate on the effect of Gardos channels in this model? Shouldn’t the opening of Ca2+-sensitive K+ channels without a change in Na+ conductance worsen volume stabilization according to Fig 4?

Reviewer #2: ≤

The paper by Ataullakhanov et al. describes the important role of the Na-pump as well as the transmembrane sodium and potassium gradients for volume stabilization at increased cation permeability. The obtained results are based on mathematical modelling. In addition to former investigations, the authors took into consideration glycolytic metabolites and adenine nucleotides. The paper is well written and the results are of importance for fundamental red blood cell research.

However, the paper would improve by taking into consideration the following points:

General:

The authors state that the red blood cells change their shape when passing through narrow capillaries. Most mammalian red blood cells can transform from discocytes to echinocytes or stomatocytes. However, the red cells from camels are ellipsoid and do not change the shape. This should be added at least in parenthesis.

In the text (Results) a permeability increase of 5 times is mentioned, whereas in Fig. 2 a 4-fold increase is given in the legend.

It is mentioned that a sodium and potassium gradient is existing in most mammalian cells. Investigating red blood cells, it should be mentioned that in most cases this is based on the sodium pump. However, dog and cat red cells do not have a sodium pump but have also such gradients. It is due to the existence of a Ca/Na exchanger (absent in other mammalian red blood cells), which produces based on the calcium pump a Na gradient and based on this Na gradient a Na,K,2Cl cotransporter produces a K gradient.

I would suggest to take out the statement given in lines 295 – 299. It would need more basic research on membrane ion transport processes.

Furthermore, the authors take into consideration the Gardos channel as an additional ion transport system. However, there are much more cation transport systems in the red blood cell membrane. I would like to have a paragraph mentioning these systems and discussing a possible role in the effect investigated in the present paper. At least the authors should say that it is planned to take such ion transport systems into account.

Minor points:

Throughout the paper “erythrocyte” or “red blood cells” should be used. In addition, non-selective should be written with – throughout the paper. Define A in mathematical equations.

Line 28: the role …

Line 105: Instead: … reduced significantly more. Better: … reduced to a greater extend.

Line 122: … why an erythrocyte …

Line 124: Delete space in Na/ K-ATPase

Line 127: … of an erythrocyte.

Line 131: Instead of disturbance – change of the permeability

Line 132: Instead of undergoes – underlie significant mechanical stress, ….

Lines 156, 348, 355 – comma before respectively.

Lines 202, 234, 240 - Figs.

Line 287: Instead of … damage to the cell membrane of these cells - … effect on the membrane of these cells ….

Line 335: Comma before which

Line 357 (and further on, e.g. 438): There is no °K – only K.

Line 418: 5.5

Line 429: ATP not Russian ATF

Table: 3 times parameter in Russian.

References: Instead of brackets it should be stated (in Russian) – ref 3, 26, 33, 38.

What means n.d. in ref. 3?

Some references (titles) are written with capital letters (20, 32, 34, 36, 39). Most references, however with small letters.

6. PLOS authors have the option to publish the peer review history of their article (what does this mean?). If published, this will include your full peer review and any attached files.

Reviewer #1: No

Reviewer #2: **Yes: **Prof. Dr. Ingolf Bernhardt

---

## [Author Response · Author response to Decision Letter 0]

8 Nov 2022

We are grateful to the Reviewers for their valuable comments and suggestions which, to our opinion, caused a significant improvement of the manuscript. Below are our responses to the Reviewer’s comments.

Reviewer #1

The paper by Ataullakhanov et al entitled “Significance of two transmembrane ion gradients for human erythrocyte volume stabilization” is a mathematical modeling study of the role of ionic gradients in stabilization of erythrocyte volume. Maintenance of an optimal surface-to-volume ratio is vital for the physiological function of erythrocytes as it allows them to deform easily inside the capillaries that are several-fold smaller in diameter than the cell if it were a sphere. This manuscript focuses on the benefits of having two (Na+/K+) transmembrane gradients instead of just one (Na+) and demonstrates that larger dynamic range of Na+ concentrations provided by the activity of Na+/K+-ATPase is critical for RBC volume stabilization under several-fold increases in non-specific membrane permeability to ions. The study is very interesting and has applications for other cells, such as neurons where large dynamic range of transmembrane ionic gradients is similarly physiologically vital. I have several suggestions that in my opinion will improve the manuscript and make it more accessible for wider audiences.

Major:

1. While the model describes transmembrane osmotic gradients, electrical component of ionic gradients is not discussed in the main text. The model does incorporate an algebraic equation for electroneutrality inside the cell and the dependence of passive ion fluxes on membrane potential, but it appears that erythrocyte membrane potential is a constant rather than the variable, which is a significant limitation of this study that the authors should discuss.

The erythrocyte transmembrane potential is not a constant in the model. It is associated with the distribution of Cl- ions across the cell membrane because the cell membrane permeability for Cl-anions in the erythrocyte is two orders of magnitude higher than for cations [1]. In the presence of two transmembrane ion gradients an increase in cell membrane permeability to cations causes a redistribution of Na+ and K+ ions across the cell membrane while the distribution of Cl- anions and transmembrane potential almost do not change. However, in the case of only one transmembrane ion gradient the transmembrane potential changes dramatically with the change in the cell membrane permeability to cations. We added corresponding explanations and figure illustrating the behavior of transmembrane potential to the manuscript (lines 273-290, Fig. 5).

2. As the focus of the manuscript is on S/V stabilization, I am curious to see how robust the model is to account for natural cell-to-cell variability. For example, how well is S/V maintained when Na+/K+-ATPase surface density is changed or when intracellular “passive” osmotic pressure is varied?

Variation of most model parameters do not affect stabilization of erythrocyte cell volume significantly. We added a new paragraph (lines 291-311) and a figure (Fig. 6) that show that the cell volume is stabilized even if Na/K-ATPase or hexokinase activity increase or decrease two times compared with the normal value. However, the erythrocyte cell volume is not stabilized and changes proportional to a content of impermeable molecules in the cell or proportional to osmolarity of the medium that confirms the reputation of the human erythrocyte as an ideal osmometer (lines 312-316, Fig. 7).

Minor:

1. I don’t see much added value in the 3rd model (Na+/K+-ATPase without Na+-mediated activation). The authors should consider moving these data to supplementary materials or removing them from the manuscript.

The main role of this model is to demonstrate that the volume stabilization in the case of one transmembrane ion gradient with the regulated transport Na-ATPase is almost as bad as with unregulated Na-ATPase. We added this explanation to the text (lines 200-204) and prefer to keep the 3-d model version in the main manuscript.

2. Can the authors speculate on the effect of Gardos channels in this model? Shouldn’t the opening of Ca2+-sensitive K+ channels without a change in Na+ conductance worsens volume stabilization according to Fig 4?

We studied a possible effect of Gardos channels on human erythrocyte volume stabilization in the paper published in the Biophysical Chemistry, 80, 199-215, 1999 [2]. Of course, opening of Gardos channels along can worsen the stabilization of human erythrocytes cell volume. And it happens probably in case of pathology named hereditary xerocytosis [3]. However, it was shown that in case of increase in non-selective cell membrane permeability the functioning of the Gardos channels can significantly improve the erythrocyte cell volume stabilization [2] (see lines 380-384 in Discussion).

References

1. Vitvitsky VM, Frolova E V, Martinov M V, Komarova S V, Ataullakhanov FI. Anion permeability and erythrocyte swelling. Bioelectrochemistry 2000;52:169–77.

2. Martinov M V., Vitvitsky VM, Ataullakhanov FI. Volume stabilization in human erythrocytes: Combined effects of Ca2+-dependent potassium channels and adenylate metabolism. Biophys Chem 1999; 80:199–215.

3. Gallagher P.G. Disorders of erythrocyte hydration. Blood 130, 2699-2708, 2017.

Reviewer #2

The paper by Ataullakhanov et al. describes the important role of the Na-pump as well as the transmembrane sodium and potassium gradients for volume stabilization at increased cation permeability. The obtained results are based on mathematical modelling. In addition to former investigations, the authors took into consideration glycolytic metabolites and adenine nucleotides. The paper is well written and the results are of importance for fundamental red blood cell research.

However, the paper would improve by taking into consideration the following points:

General:

The authors state that the red blood cells change their shape when passing through narrow capillaries. Most mammalian red blood cells can transform from discocytes to echinocytes or stomatocytes. However, the red cells from camels are ellipsoid and do not change the shape. This should be added at least in parenthesis.

Thank you for this correction. We mentioned this information in the Introduction (lines 57-67) with corresponding references.

In the text (Results) a permeability increase of 5 times is mentioned, whereas in Fig. 2 a 4-fold increase is given in the legend.

We note in the text (lines 172-175) that at 5-fold increase in cell membrane permeability erythrocyte gets a spherical shape and any further increase in the permeability causes the cell damage. In the Fig. 2 we demonstrate a kinetics of cell volume and intracellular [Na+] at cell membrane permeability increase (4-fold) that does not cause the cell destruction yet. In the revised version of the manuscript, we indicated limits of erythrocyte volume increase in Fig 2A and in new figures (Fig 6, and Fig 7). 

It is mentioned that a sodium and potassium gradient is existing in most mammalian cells. Investigating red blood cells, it should be mentioned that in most cases this is based on the sodium pump. However, dog and cat red cells do not have a sodium pump but have also such gradients. It is due to the existence of a Ca/Na exchanger (absent in other mammalian red blood cells), which produces based on the calcium pump a Na gradient and based on this Na gradient a Na, K,2Cl cotransporter produces a K gradient.

We know about an “unusual” mechanism of the maintenance of transmembrane sodium ion gradient in dog and cat erythrocytes. We think that it would be very interesting to investigate how effective this mechanism is in maintaining a constant surface area to volume ratio. But this mechanism is not well studied to our knowledge. Also, it was shown recently, that at least in some dog breeds animals with normal transport Na/K-ATPase in erythrocytes exist, that maintain Na+ and K+ transmembrane gradients similar to other species [1-3]. This topic is beyond of aims of our study and we would not like to discuss it here.

I would suggest to take out the statement given in lines 295 – 299. It would need more basic research on membrane ion transport processes.

To our knowledge, there are no good rational hypotheses regarding necessity of two opposite transmembrane ion gradients with high intracellular concentration of K+ ions in majority of cells. We do not pretend to present a complete explanation of that phenomenon but, in our opinion, we offer a reasonable assumption about it and would like to mention it in the text.

Furthermore, the authors take into consideration the Gardos channel as an additional ion transport system. However, there are much more cation transport systems in the red blood cell membrane. I would like to have a paragraph mentioning these systems and discussing a possible role in the effect investigated in the present paper. At least the authors should say that it is planned to take such ion transport systems into account.

Thank you for this comment. We mentioned the cation transport systems presented in human erythrocytes in the Introduction (lines 120-122). Also, we mentioned in the Discussion section (line 384-386) the necessity to investigate a contribution of other ion transport system in erythrocyte volume stabilization.

Minor points:

Throughout the paper “erythrocyte” or “red blood cells” should be used (Corrected). In addition, non-selective should be written with – throughout the paper (Corrected). Define A in mathematical equations – A description of parameter A was added to the text (lines 435, 436).

Line 28: the role …- Corrected

Line 105: Instead: … reduced significantly more. Better: … reduced to a greater extend. - Corrected

Line 122: … why an erythrocyte …- Corrected

Line 124: Delete space in Na/ K-ATPase - Corrected

Line 127: … of an erythrocyte. - Corrected

Line 131: Instead of disturbance – change of the permeability - Corrected

Line 132: Instead of undergoes – underlie significant mechanical stress, ….-Corrected

Lines 156, 348, 355 – comma before respectively.- Corrected

Lines 202, 234, 240 - Figs. - Corrected

Line 287: Instead of … damage to the cell membrane of these cells - … effect on the membrane of these cells ….- We corrected the phrase in a different way (Line 350).

Line 335: Comma before which - Corrected

Line 357 (and further on, e.g. 438): There is no °K – only K. - Corrected

Line 418: 5.5 - Corrected

Line 429: ATP not Russian ATF (actually, it was line 492) - Corrected

Table: 3 times parameter in Russian. - Corrected

References: Instead of brackets it should be stated (in Russian) – ref 3, 26, 33, 38.- the brackets were added by the reference manager software (Mendeley). We stated that the referred papers are in Russian.

What means n.d. in ref. 3?

Some references (titles) are written with capital letters (20, 32, 34, 36, 39). Most references, however with small letters. – All issues regarding the References were corrected.

References

1. Maede Y., Inaba M., Taniguchi N. Increase of Na-K-ATPase activity, glutamate, and aspartate uptake in dog erythrocytes associated with hereditary high accumulation of GSH, glutamate, glutamine, and aspartate. Blood 61, 493-499, 1983.

2. Fujise H., Higa K., Nakayama T., Wada K., Ochiai H., Tanabe Y. Incidence of dog possessing red blood cell with high K in Japan and East Asia. J. Vet. Med. Sci. 59, 497-497, 1997.

3. Conrado F.O., Oliveira S.T., Lacerda L.A., Silva M.O.D., Hlavac N., Gonzalez F.H.D. Clinicopathologic and electrocardiographic features of Akita dogs with high and low erythrocyte potassium phenotypes. Vet. Clin. Pathol. 43, 50–54, 2014.

Sincerely yours

Victor M. Vitvitsky, D.Sc.

Center for Theoretical Problems of Physico-Chemical Pharmacology

Russian Academy of Sciences

Moscow 109029, Russia

E-mail: victor_vitvitsky@yahoo.com

---

## [Decision Letter · Decision Letter 1]

6 Dec 2022

Significance of two transmembrane ion gradients for human erythrocyte volume stabilization.

PONE-D-22-20774R1

Dear Dr. Vitvitsky,

We’re pleased to inform you that your manuscript has been judged scientifically suitable for publication and will be formally accepted for publication once it meets all outstanding technical requirements.

Kind regards,

Pramod K. Yadav, Ph. D.

Academic Editor

PLOS ONE

Additional Editor Comments:

Suggestions to the authors by Dr. Ingolf Bernhardt need to be incorporated

The manuscript improved significantly after revision. However, there are some points to take into consideration.

Please go through the manuscript again and change Na+ ions and sodium ions in Na+ only. It should be identical everywhere (Na+ is an ion, you do not need to write ion. Alternatively Na ions). In some places it is correct!

Line 15: Do not start a sentence with „And …..“.

Line 270: version 2 (small v), cp. other places

Line 297: Point is missing after Fig

Line 394: Why here capital letters (see below other headlines)

Reviewers' comments:

Reviewer's Responses to Questions

**Comments to the Author**

1. If the authors have adequately addressed your comments raised in a previous round of review and you feel that this manuscript is now acceptable for publication, you may indicate that here to bypass the “Comments to the Author” section, enter your conflict of interest statement in the “Confidential to Editor” section, and submit your "Accept" recommendation.

Reviewer #1: All comments have been addressed

Reviewer #2: (No Response)

2. Is the manuscript technically sound, and do the data support the conclusions?

Reviewer #1: Yes

Reviewer #2: Yes

3. Has the statistical analysis been performed appropriately and rigorously? 

Reviewer #1: N/A

Reviewer #2: Yes

4. Have the authors made all data underlying the findings in their manuscript fully available?

Reviewer #1: Yes

Reviewer #2: Yes

5. Is the manuscript presented in an intelligible fashion and written in standard English?

Reviewer #1: Yes

Reviewer #2: Yes

6. Review Comments to the Author

Reviewer #1: (No Response)

Reviewer #2: Suggestions to the authors

The manuscript improved significantly after revision. However, there are some points to take into consideration.

Please go through the manuscript again and change Na+ ions and sodium ions in Na+ only. It should be identical everywhere (Na+ is an ion, you do not need to write ion. Alternatively Na ions). In some places it is correct!

Line 15: Do not start a sentence with „And …..“.

Line 270: version 2 (small v), cp. other places

Line 297: Point is missing after Fig

Line 394: Why here capital letters (see below other headlines)

7. PLOS authors have the option to publish the peer review history of their article (what does this mean?). If published, this will include your full peer review and any attached files.

Reviewer #1: No

Reviewer #2: **Yes: **Prof. Dr. Ingolf Bernhardt

---

## [Editor Report · Acceptance letter]

12 Dec 2022

PONE-D-22-20774R1 

Significance of two transmembrane ion gradients for human erythrocyte volume stabilization. 

Dear Dr. Vitvitsky:

I'm pleased to inform you that your manuscript has been deemed suitable for publication in PLOS ONE. Congratulations! Your manuscript is now with our production department. 

Kind regards, 

on behalf of

Dr. Pramod K. Yadav 

Academic Editor

PLOS ONE